# Chemical Compositions and Source Analysis of PM2.5 during Autumn and Winter in a Heavily Polluted City in China

**Shasha Tian [1], Yingying Liu [1], Jing Wang [1], Jian Wang [1], Lujian Hou [2], Bo Lv [2], Xinhua Wang [1], Xueyan Zhao [1], Wen Yang [1], Chunmei Geng [1,\*], Bin Han [1,\*]** 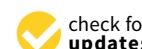 **and Zhipeng Bai [1]**

1  State Key Laboratory of Environmental Criteria and Risk Assessment, Chinese Research Academy of Environmental Sciences, Beijing 100012, China; tianss1020@163.com (S.T.); liuyy@craes.org.cn (Y.L.); wangjing@craes.org.cn (J.W.); wangjian@craes.org.cn (J.W.); wangxh@craes.org.cn (X.W.); zhaoxy@craes.org.cn (X.Z.); yangwen@craes.org.cn (W.Y.); baizp@craes.org.cn (Z.B.)
2  Jinan Environmental Monitoring Center Station, Jinan 250000, China; houlujian@vip.sina.com (L.H.); lvbo0531@163.com (B.L.)
\*  Correspondence: gengcm@craes.org.cn (C.G.); hanbin@craes.org.cn (B.H.); Tel.: +86-10-849-15246 (C.G.); +86-10-8493-5950 (B.H.)

**Abstract:** As one of the biggest cities in North China, Jinan has been suffering heavy air pollution in recent decades. To better characterize the ambient particulate matter in Jinan during heavy pollution periods, we collected daily $PM_{2.5}$ (particulate matter with aerodynamic diameters equal to or less than 2.5 μm) filter samples from 15 October 2017 to 31 January 2018 and analyzed their chemical compositions (including inorganic water-soluble ions (WSIs), carbonaceous species, and inorganic elements). The daily average concentration of $PM_{2.5}$ was 83.5 μg/m$^3$ during the sampling period. A meteorological analysis revealed that both low wind speed and high relative humidity facilitated the occurrence of high $PM_{2.5}$ pollution episodes. A chemical analysis indicated that high concentrations of water-soluble ions, carbonaceous species, and elements were observed during heavy pollution days. The major constituents of $PM_{2.5}$ in Jinan were secondary aerosol particles and organic matter based on the results of mass closure. Chemical Mass Balance (CMB) was used to track possible sources and identified that nitrate, sulfate, vehicle exhaust and coal fly ash were the main contributors to $PM_{2.5}$ during heavy pollution days in Jinan, accounting for 25.4%, 18.6%, 18.2%, and 13.3%, respectively.

**Keywords:** $PM_{2.5}$; chemical components; mass closure; chemical mass balance; Jinan

## 1. Introduction

With the rapid development of industrialization and urbanization, the Chinese economy has progressed quickly over the last few decades. However, the economic boom has imposed tremendous pressure on the environment, especially the ambient atmosphere in Northern China [1–3]. The frequent severe haze episodes after the year 2010 have aroused worldwide concern. $PM_{2.5}$ (particulate matter with aerodynamic diameters equal to or less than 2.5 μm) is regarded as the main contributor to heavy air pollution [4,5]. A previous study showed that cardiovascular morbidity and mortality are associated with long-term exposure to high $PM_{2.5}$ [6]. Another study found that 3.45 million premature deaths around the world were linked to $PM_{2.5}$ pollution in 2007 [7]. $PM_{2.5}$ was the primary air pollutant in 80% of Chinese cities in 2016 [8].

To date, the chemical compositions, potential sources, and formation mechanisms of $PM_{2.5}$ have been widely investigated in economically developed and densely populated areas [9–15]. However, the mass concentration and chemical composition characteristics of $PM_{2.5}$ vary greatly between different

areas and periods. Therefore, comprehensive studies are necessary to characterize $PM_{2.5}$ at different locations and in different periods.

Jinan is the capital city of Shandong province, and a typical industrial city in the middle eastern area of China. Jinan is a medium-sized city, with a total residential area of 10,244 km$^2$ and a population of approximately 7.4 million. Mount Tai, Mount Lu, and the Yellow River surround the city and make it a semi-enclosed area. The city has a typical continental monsoon climate, characterized by a hot and rainy summer and a cold and dry winter. The mean annual precipitation in the area is 685 mm. Coal is the main fuel in Jinan and is widely used for industry and daily life, especially for domestic heating during November to March. The number of motor vehicles exceeded 2.06 million in 2017, with an increase of 13.12% over the previous year. Specifically, low-quality fuel with high sulfur content greatly increased $SO_2$ emissions (which can lead to the secondary $PM_{2.5}$ formation) [16]. The emissions from coal combustion and vehicles are believed to be the major contributors to ambient $PM_{2.5}$. Moreover, poor natural dispersing conditions, rapid population growth, and quick economic development have also contributed to serious air pollution in Jinan. The annual average concentration of $PM_{2.5}$ at urban sites in Jinan was 148.71 μg/m$^3$ in 2006, which is one of the highest levels reported in the world [17]. The percentage of seriously hazy days in Jinan (with visibility less than 5 km) reached 8.5% in 2006 [18]. In 2012, the city ranked as one of the ten most air polluted cities around the world [19].

The size-distribution of fine particles [20,21], water-soluble ions [17,22], long-term variations of $PM_{2.5}$ concentration levels, and their chemical constituents have been widely investigated in Jinan [19]. However, our understanding of the chemical composition of $PM_{2.5}$ at different pollution levels is still incomplete, especially the driving factors in the formation of heavy pollution. In this study, we collected $PM_{2.5}$ daily samples from 15 October 2017 to 31 January 2018 at four different functional areas and analyzed their chemical compositions, including inorganic water-soluble ions (WSIs), organic/elemental carbon (OC/EC), and inorganic elements. The main objectives of this study are to illustrate the characterizations and sources of $PM_{2.5}$ under the different pollution levels during autumn and winter in Jinan. The aims of this study are to (1) characterize the chemical compositions of $PM_{2.5}$ (inorganic elements, carbonaceous species, and inorganic water-soluble ions) in Jinan and (2) quantitatively assess the source contributions of $PM_{2.5}$ by using a chemical mass balance (CMB) model and chemical mass closure. The results gained from this study will help us better understand the pollution characteristics of $PM_{2.5}$ and recognize the potential sources of $PM_{2.5}$ during different polluted days in Jinan. Importantly, these findings will help the local government make future air pollution control policies, especially emergent countermeasures against heavy pollution events.

## 2. Materials and Methods

### 2.1. $PM_{2.5}$ Samples

$PM_{2.5}$ samples were simultaneously collected in four different functional areas of Jinan (Figure 1), and more details were shown in Table 1. The sampling equipment was placed on the roofs of the buildings (20 m above the ground). $PM_{2.5}$ samples were collected daily (from 9:00 a.m. to 8:00 a.m. the next day) with a small-flow particulate matter sampler (Commodore Measurement Technology Co., Ltd., Germany, 16.7 L/min). A total of 720 samples were collected from 15 October 2017 to 31 January 2018. $PM_{2.5}$ samples were collected on quartz filters (47 mm, Whatman, UK) and Teflon filters (47 mm, Pall, UK), which were pre-heated at 450 °C for 4h before use in order to reduce the residual carbon levels that were associated with the filters. The filters were equilibrated under a constant temperature (20 ± 1 °C) and humidity (50 ± 5%) over 48 h. The filters were individually wrapped in aluminum foil and were sealed in polyethylene bags and stored in a fridge at −20 °C before analysis.

**Table 1.** Sampling site statistics in Jinan.

| Sites | Longitude and Latitude | Functional Areas |
|---|---|---|
| Environmental Monitoring Station | 116°57′ E, 36°39′ N | city downtown area |
| Architectural University | 11710′ E, 36°40′ N | New urban area |
| Lanxiang Technical School | 116°56′ E, 36°42′ N | Suburban junction area |
| Jigang | 117°10′ E, 36°43′ N | Industrial area |

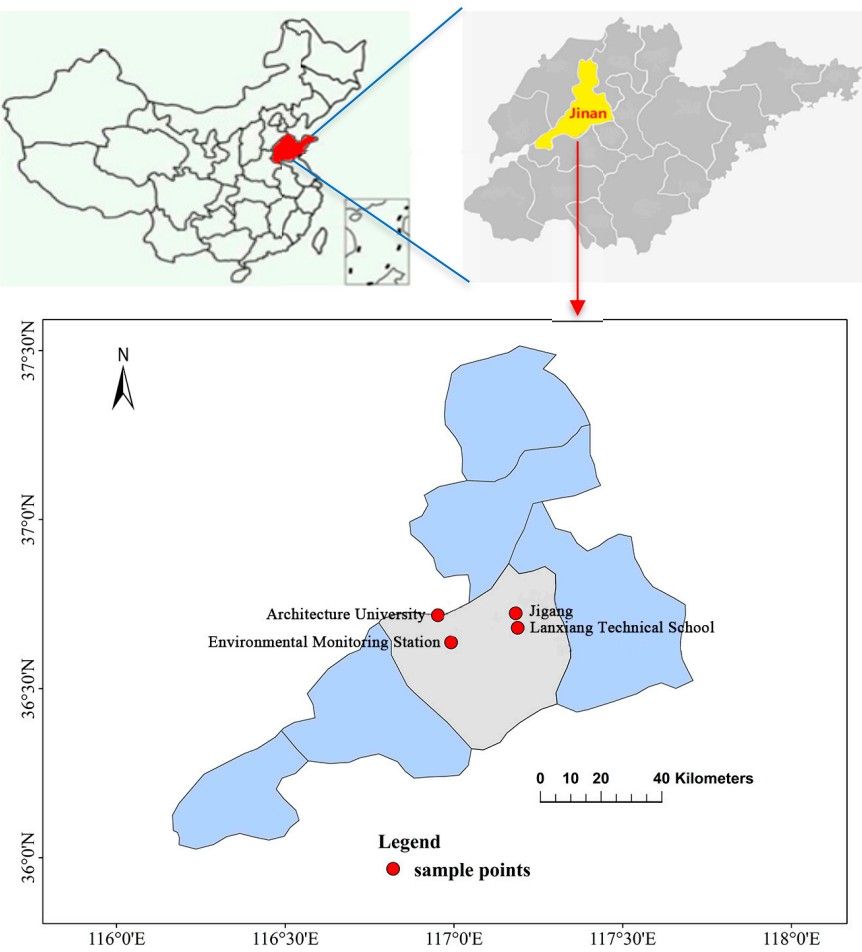

**Figure 1.** The four sampling sites in Jinan, China.

## *2.2. Chemical Analysis*

Both the Teflon and quartz filters were prebaked before use to remove any absorbed organic material. The samples and blank filters were weighed before and after sampling at least three times using a millionth automatic balance (Commodore AWS-1, German) with a precision of 0.04 mg, and the differences in the filter weights were divided by the sampling volume in order to calculate the $PM_{2.5}$ mass concentrations.

One fourth of each quartz filter sample was put into the sampling tube, and 10 mL of ultrapure water was added, ultrasonicated for 20 min, and centrifugalized for 10 min. Then, the extract was filtered with a 0.45 μm filter membrane to remove any insoluble species. Finally, the supernatant solution was taken into a clean bottle to analyze water-soluble ions. Dionex Inc., ICS-2000 was used to analyze the cations ($Na^+$, $NH_4^+$, $K^+$, $Mg^{2+}$ and $Ca^{2+}$) and ICS-3000 for anions ($Cl^-$, $NO_3^-$ and $SO_4^{2-}$). The detection limits of these water-soluble ions ranged from 0.005 to 0.085 μg/m$^3$, and the precisions were <10% [23,24].

The carbonaceous species were analyzed by DRI Model 2001 (thermal/optical carbon analyzer) for organic carbon (OC) and elemental carbon (EC), following the Interagency Monitoring of Protected Visual Environment (IMPROVE) [25]. The specific steps were as follows. We cut off a 0.5 $cm^2$ punch area from the quartz filter with a cutting knife and put it into the thermal carbon analyzer. First, nonoxidizing helium was fed into the muffle furnace, and the DRI Model 2001 was heated to 120, 250, 450 and 550 °C, which yielded OC1, OC2, OC3, and OC4. Then, the samples were heated in helium containing 2% oxygen, which produced EC1, EC2, and EC3, finally yielding OC = OC1 + OC2 + OC3 + OC4 + OPC, EC = EC1 + EC2 + EC3 - OPC, TC = OC + EC = OC1 + OC2 + OC3 + OC4 + EC1 + EC2 + EC3. The detection limits were 0.18 ± 0.06 and 0.04 ± 0.01 $\mu g/m^3$ for OC and EC, respectively.

For the element analysis, half of one Teflon filter sample was extracted by hydrofluoric acid and diluted chloroazotic acid in order to analyze Na, K, Cr, Ni, Cu, Zn, As, Mo, Cd, Sn, Sb, Tl, Pb and Bi using the inductively coupled plasma-mass spectrometry analysis method (Agilent 7500 a, Agilent Technologies, CA, USA). The other half was first ashed and then extracted by solid sodium hydroxide to analyze Al, Mg, Ca, Fe, and Si using the inductively coupled plasma optical emission spectrometry method (Optima 4300 DV, PerkinElmer, MA, USA). The method detection limit (MDL) of the elements ranged from 0.02 to 30 $ng\ m^{-3}$, and the precisions were <10% [26,27].

Meteorological parameters, such as wind speed, temperature (T), and the relative humidity (RH) were selected from the website of the Weather Underground) [28]. Hourly $SO_2$ and $NO_2$ data were obtained from [29]

### 2.3. Date Analysis

#### 2.3.1. Ionic Balance

Understanding the acidity of the PM is important since it is a key factor that affects the heterogeneous reactions, hygroscopic growth, and toxicity of aerosols [30,31]. Ion balance calculations were useful for studying the acid–base balance of the aerosol. The ratios of AE (anion equivalent) and CE (cation equivalent) were used to indicate the acidity of the atmospheric aerosol. The calculations of the anion and cation are as follows [30,32]:

$$AE = [NO_3^-]/62 + [SO_4^{2-}]/48 + [Cl^-]/35.5 \tag{1}$$

$$CE = [NH_4^+]/18 + [Ca^{2+}]/20 + [K^+]/39 + [Mg^{2+}]/12 + [Na^+]/23 \tag{2}$$

where $[NO_3^-]$, $[SO_4^{2-}]$, $[Cl^-]$, $[NH_4^+]$, $[Ca^{2+}]$, $[K^+]$, $[Mg^{2+}]$, and $[Na^+]$ represent the mass concentrations of these ion compositions in the $PM_{2.5}$ samples. Particles were in an acidic condition when the AE/CE ratio was larger than 1, while a ratio lower than 1 indicated an alkaline condition [33].

#### 2.3.2. Oxidation Ratio

Previous articles have proven that the oxidation degree of gaseous pollutants is reflected by the sulfur oxidation ratio (SOR) and the nitrogen oxidation ratio (NOR), respectively, to the extent that $SO_2$ and $NO_x$ are oxidized to form $SO_4^{2-}$ and $NO_3^-$ [34]:

$$SOR = n(SO_4^{2-})/[n(SO_4^{2-}) + n(SO_2)] \tag{3}$$

$$NOR = n(NO_3^-)/[n(NO_3^-) + n(NO_2)] \tag{4}$$

where $n(SO_4^{2-})$, $n(SO_2)$, $n(NO_3^-)$, $n(NO_2)$ represent the molar concentrations of sulfate, sulfur dioxide, nitrate, and nitrogen dioxide, respectively.

### 2.3.3. Enrichment Factor

The enrichment factor (EF) method can be used to evaluate the sources of elemental compositions in $PM_{2.5}$. The method is as follows [35–37]:

$$EF = (C_i/C_R)_{sample} / (C_i/C_R)_{crust} \qquad (5)$$

where $(C_i/C_R)_{sample}$ represents the ratio of the concentration of the targeted element to the concentration of a reference element in the aerosol; $(C_i/C_R)_{crust}$ represents the ratio of the target element to a reference element in the Earth's crust [38]. Al was chosen as a reference element due to its stability in most anthropogenic contaminants in this study. EF < 10 indicates that the EFs are mainly from natural sources; EF ≥ 10 suggests that the EFs are mostly from anthropogenic sources.

### 2.3.4. Mass Reconstruction

In this study, the $PM_{2.5}$ mass was reconstructed by organic matter (OM), mineral dust, trace elements (TE), $SO_4^{2-}$, $NO_3^-$, $NH_4^+$, and EC. The OM mass was calculated by the organic carbon (OC) mass concentration multiplied by 1.6 for urban areas [39], as follows:

$$OM = OC \times 1.6 \qquad (6)$$

The mineral dust composition is the sum of the mass concentration of the elemental oxides in the crust [40], as in the following equation:

$$[Mineral] = 1.89\varrho(Al) + 2.14\varrho(Si) + 1.21\varrho(K) + 1.4\varrho(Ca) + 1.66\varrho(Mg) + 1.7\varrho(Ti) + 1.43\varrho(Fe) \qquad (7)$$

TE is the sum of the mass concentrations of the trace elements other than crust and sea salt. The concentrations of $SO_4^{2-}$, $NO_3^-$, $NH_4^+$ and EC were directly measured [32].

### 2.3.5. Source Apportionment

Source contribution estimates (SCEs) were the main outputs of the chemical mass balance (CMB) model, which represents the fractional contribution to the ambient $PM_{2.5}$ concentration by each source profile used in the model [32]. The principle can be expressed as

$$C = \sum_{J-1}^{J} S_j \, j = 1,\, 2,\, \ldots J \qquad (8)$$

where C represents the total mass measured at the receptor, $\mu g/m^3$; $S_j$ represents the contribution from an individual source, $\mu g/m^3$; and j represents the number of sources. The concentration of elemental composition i, $C_i$, will be

$$C_i = \sum_{j=1}^{J} S_j \times F_{ij} \, i = 1,\, 2,\, \ldots I;\, j = 1,\, 2,\, \ldots J \qquad (9)$$

where $F_{ij}$ represents the fraction of source contribution $S_j$ composed of element i, g/g; $S_j$ is the contribution from an individual source, $\mu g/m^3$; j represents the number of sources; and i represents the number of chemical compositions.

Therefore, the contribution concentration and total contribution rate of each pollution source to the different chemical compositions in atmospheric particulate matter can be obtained as long as the composition spectrum of each pollution source and the compositional spectra of the acceptor's atmospheric particulate matter are known.

*2.4. Establishment of Source Profile*

Fugitive dust was collected on windowsills positioned at heights of 5–15 m in residential areas. The soil dust mainly originated from bare croplands, hilly country, dry riverbeds in the vicinity of Jinan, and exposed land within urban areas. Cement dust was collected from the roofs of residential buildings around the building site or from production lines of nearby cement factories. Coal combustion fly ash was collected from the stacks of different coal burning facilities by using a dilution system. Vehicle exhaust dust was obtained from 2 buses, 4 taxis, 3 small trucks, 2 medium-sized passenger cars, 2 large passenger vehicles, 15 small passenger vehicles, and 2 large-sized trucks. Iron making dust was obtained from the industrial furnaces of top companies (utilizing different processes) with the highest total pollutant emissions in Jinan. Biomass emission samples were collected from the biomass combustion boiler. The source profiles of secondary sulfate and nitrate used the constitutions of $(NH4)_2SO_4$ and $NH_4NO_3$, respectively. All of the sampling methods and source profile constructions followed the Technical Guideline of Source Apportionment for Particulate Matter (Chinese version available [41]).

## 3. Results and Discussions

*3.1. The Characteristics of $PM_{2.5}$ and Chemical Compositions*

### 3.1.1. $PM_{2.5}$ Mass Concentration

The trends of $PM_{2.5}$ mass concentrations in the four different functional areas of Jinan were relatively similar during the sampling period. Therefore, the data of the four sites were pooled together for discussion to more objectively represent the pollution status of Jinan. The time-series data of the concentrations of $PM_{2.5}$ and meteorological parameters are shown in Figure 2. The mass concentration of $PM_{2.5}$, water-soluble inorganic ions (WSIs), inorganic elements, OC, and EC at different pollution levels during the sampling period are listed in Table 2. In this study, the mass concentrations of $PM_{2.5}$ ranged from 20.7 $\mu g/m^3$ (11 January) to 237.2 $\mu g/m^3$ (17 January), with an average of 83.5 (Standard deviaton: 49.59) $\mu g/m^3$, which is higher than the secondary limit of the National Ambient Air Quality Standard of $PM_{2.5}$ (75 $\mu g/m^3$). As shown in Table S1, the mass concentrations of $PM_{2.5}$ were found to be lower than those in north China, such as Lanzhou [12], Xi'an [42], Beijing [43], as well as one previous study on Jinan [19], but higher than those in south cities, such as Nanjing [44].

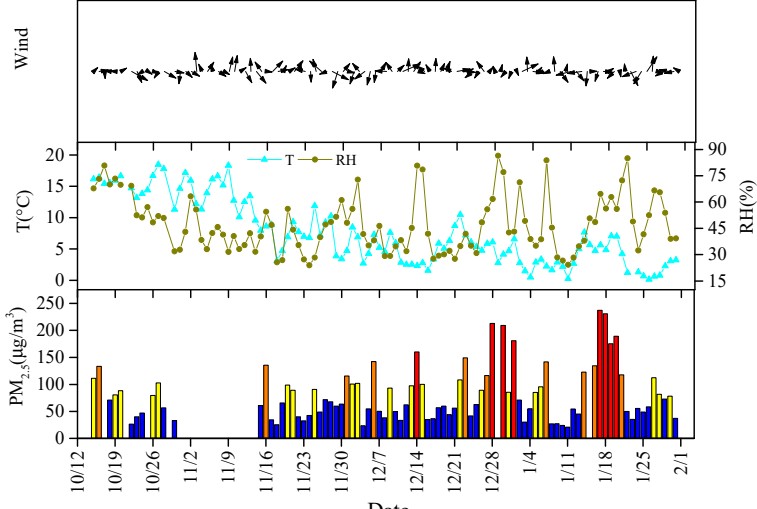

**Figure 2.** Time series of daily $PM_{2.5}$ and meteorological date (blue column represents good days, yellow column represents light days, orange column represents moderate days, red column represents heavy days).

**Table 2.** Mass concentration of major compositions in $PM_{2.5}$ in different pollution days.

| Compositions | Good | Light | Moderate | Heavy |
|---|---|---|---|---|
| $F^-$ ($\mu g/m^3$) | 0.1 ± 0.06 | 0.1 ± 0.07 | 0.1 ± 0.16 | 0.1 ± 0.10 |
| $Cl^-$ ($\mu g/m^3$) | 2.0 ± 1.04 | 3.6 ± 1.78 | 4.1 ± 2.34 | 6.7 ± 1.97 |
| $NO_3^-$ ($\mu g/m^3$) | 8.2 ± 4.47 | 22.2 ± 6.21 | 31.2 ± 5.01 | 47.2 ± 10.44 |
| $SO_4^{2-}$ ($\mu g/m^3$) | 4.7 ± 2.56 | 10.6 ± 3.27 | 14.3 ± 4.47 | 27.8 ± 4.77 |
| $NH_4^+$ ($\mu g/m^3$) | 4.7 ± 2.06 | 10.2 ± 1.95 | 14.2 ± 2.39 | 23.9 ± 3.51 |
| $Na^+$ ($\mu g/m^3$) | 0.4 ± 0.21 | 0.4 ± 0.21 | 0.5 ± 0.21 | 0.7 ± 0.20 |
| $K^+$ ($\mu g/m^3$) | 0.5 ± 0.18 | 0.9 ± 0.30 | 1.2 ± 0.38 | 1.9 ± 0.36 |
| $Mg^{2+}$ ($\mu g/m^3$) | 0.1 ± 0.02 | 0.1 ± 0.02 | 0.1 ± 0.03 | 0.1 ± 0.05 |
| $Ca^{2+}$ ($\mu g/m^3$) | 0.9 ± 0.26 | 1.0 ± 0.39 | 1.2 ± 0.46 | 1.1 ± 0.50 |
| OC ($\mu g/m^3$) | 7.8 ± 2.05 | 12.7 ± 2.59 | 17.8 ± 3.05 | 25.9 ± 4.03 |
| EC ($\mu g/m^3$) | 3.4 ± 1.17 | 5.5 ± 2.00 | 7.2 ± 2.11 | 10.1 ± 1.80 |
| OC/EC | 2.3 ± 0.76 | 2.3 ± 1.08 | 2.5 ± 1.04 | 2.6 ± 0.26 |
| Na ($ng/m^3$) | 452.1 ± 213.65 | 496.0 ± 218.98 | 838.8 ± 320.93 | 931.4 ± 243.60 |
| Al ($ng/m^3$) | 1107.4 ± 461.19 | 1189.8 ± 556.57 | 1368.3 ± 490.82 | 1790.9 ± 897.08 |
| Si ($ng/m^3$) | 978.7 ± 396.31 | 1050.5 ± 420.50 | 1257.9 ± 688.98 | 1215.4 ± 441.30 |
| K ($ng/m^3$) | 713.5 ± 260.28 | 1069.0 ± 403.55 | 1704.7 ± 475.89 | 2117.9 ± 429.00 |
| Cr ($ng/m^3$) | 17.1 ± 21.41 | 17.4 ± 15.91 | 29.9 ± 25.43 | 18.8 ± 4.90 |
| Fe ($ng/m^3$) | 752.9 ± 235.87 | 867.1 ± 287.09 | 1161.9 ± 408.66 | 1343.1 ± 462.69 |
| Ni ($ng/m^3$) | 10.3 ± 23.22 | 7.5 ± 16.17 | 13.0 ± 18.43 | 5.8 ± 0.85 |
| Cu ($ng/m^3$) | 14.7 ± 4.26 | 20.3 ± 6.42 | 27.9 ± 8.83 | 28.2 ± 3.83 |
| Zn ($ng/m^3$) | 118.9 ± 54.08 | 159.8 ± 70.78 | 255.3 ± 67.33 | 313.4 ± 81.75 |
| As ($ng/m^3$) | 5.6 ± 2.88 | 8.0 ± 2.80 | 8.3 ± 2.10 | 12.4 ± 1.37 |
| Mo ($ng/m^3$) | 0.4 ± 0.31 | 0.8 ± 0.68 | 1.1 ± 0.56 | 1.7 ± 0.39 |
| Cd ($ng/m^3$) | 1.2 ± 0.93 | 1.9 ± 0.90 | 4.0 ± 2.19 | 3.8 ± 1.32 |
| Sn ($ng/m^3$) | 2.4 ± 1.68 | 3.6 ± 2.74 | 6.1 ± 3.39 | 9.8 ± 1.17 |
| Sb ($ng/m^3$) | 2.3 ± 0.26 | 3.9 ± 3.16 | 5.9 ± 3.12 | 9.1 ± 1.53 |
| W ($ng/m^3$) | 0.3 ± 0.26 | 0.5 ± 0.37 | 0.6 ± 0.33 | 2.0 ± 0.31 |
| Tl ($ng/m^3$) | 0.3 ± 0.22 | 0.5 ± 0.35 | 0.8 ± 0.32 | 1.1 ± 0.21 |
| Pb ($ng/m^3$) | 38.1 ± 14.73 | 63.4 ± 21.60 | 91.7 ± 27.09 | 103.3 ± 10.06 |
| Bi ($ng/m^3$) | 0.7 ± 0.42 | 1.0 ± 0.71 | 1.7 ±0.83 | 2.6 ± 0.55 |

In this study, we divided the mass concentrations of $PM_{2.5}$ into four grades according to the Technical Regulations of the Environmental Air Quality Index (AQI) (HJ 633–2012): good (≤75 $\mu g/m^3$), light (75–115 $\mu g/m^3$), moderate (115–150 $\mu g/m^3$), and heavy (≥150 $\mu g/m^3$). During the sampling period, the mass concentrations of $PM_{2.5}$ during heavy pollution days was approximately 4.3, 2.1, and 1.5 times higher than those during good, light, and moderate pollution days, respectively. The average wind speed during the sampling period was 1.3 m/s, with the highest speed observed on good days and the lowest observed on heavy pollution days. In contrast, the relative humidity followed the order: heavy pollution days > moderate pollution days > light pollution days > good days. The stable atmosphere and low mixed boundary layer height, which frequently occur when the wind speed is low, are adverse to the diffusion of air pollutants and consequently lead to the accumulation of air pollutants. Furthermore, high relative humidity accelerates the formation of new particles and the growth of aerosol particles, which results in the aggravation of the pollution level in the atmosphere [45].

### 3.1.2. Water-Soluble Ions

WSIs are important components of atmospheric particles. Figure 3 shows the mass concentrations of the major chemical compositions of $PM_{2.5}$ during the sampling period. The mass concentrations of WSIs accounted for 43.5%, 50.6%, 52.1%, and 55.1% of $PM_{2.5}$ under good, light, moderate, and heavy pollution levels, respectively. The mass concentrations of the WSIs in the four pollution levels followed the same order: $NO_3^- > SO_4^{2-} > NH_4^+ > Cl^- > K^+ > Ca^{2+} > Na^+ > Mg^{2+}$. SNA ($NO_3^-$, $SO_4^{2-}$, and $NH_4^+$) were the dominant components, accounting for 81.8%, 87.5%, 89.3%, and 90.3% of the total WSI concentrations in the four pollution levels. The high concentrations of SNA on polluted days indicated

the high formation of secondary inorganic aerosol. Respectively, the mean concentrations of $NO_3^-$, $SO_4^{2-}$, and $NH_4^+$ contributed 21.2%, 11.6%, and 10.6% to $PM_{2.5}$. Compared with other cities, the levels of $NO_3^-$ and $NH_4^+$ were higher than those in Xinjiang [45], Shanghai [11], and Chongqing [46], but the concentrations of $SO_4^{2-}$ were lower than those in the above cities. Other WSIs ($Na^+$, $K^+$ and $Cl^-$) were observed to be higher in concentration during polluted days, likely due to industrial emissions, coal combustion, or biomass burning. The concentrations of $Ca^{2+}$ and $Mg^{2+}$ were higher on moderate and heavy pollution days and mainly came from soil dust, road dust, and construction dust [22,47,48].

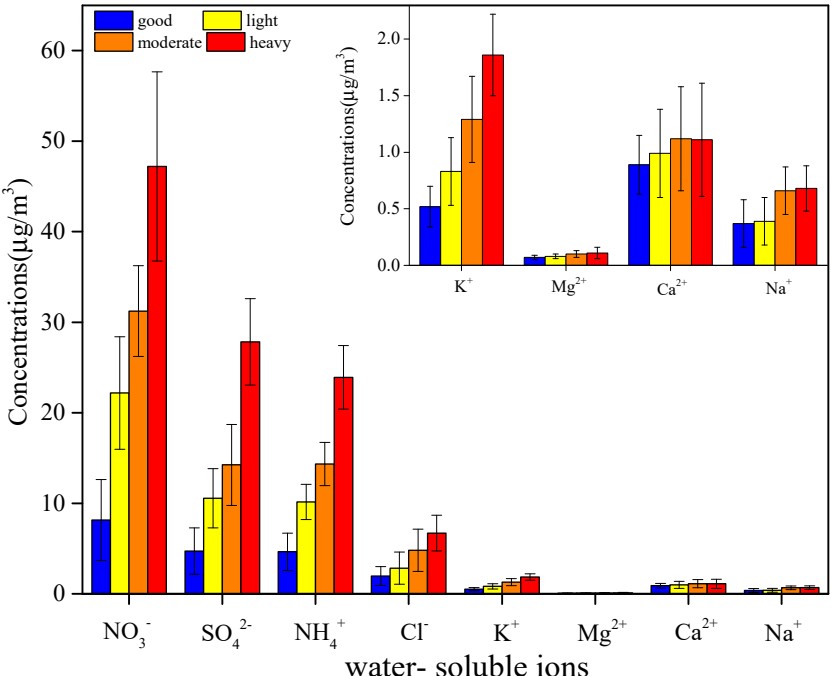

**Figure 3.** The concentrations of water–element compositions in four pollution days.

The relationships of SOR/NOR to temperature/RH during the sampling period are shown in Figure 4. The original sulfate and nitrate can absorb water and enlarge the surface area of the particles with an increasing RH, which favors the heterogeneous reactions of $NO_3^-$ and $SO_4^{2-}$, thereby increasing the SOR and NOR [49]. However, the rising trend of SOR was more obvious than that of NOR, especially when the RH was greater than 60%. At the same time, the Spearman correlation coefficients of SOR and NOR with relative humidity were 0.7 and 0.5, respectively, which also indicated that the influence of relative humidity on SOR was greater than that on NOR. Moreover, SOR showed a weaker correlation with temperature, and NOR decreased with an increase in temperature, which is related to the volatilization of $NH_4NO_3$ at higher temperatures [50].

The anion and cation equivalents discussed in this study have strong correlations with the four pollution levels, as well as the major ions ($NH_4^+$ and $NO_3^- + SO_4^{2-}$) (Figures 5 and 6), demonstrating the reliability of the analytical results. The mean equivalent ratios of AE/CE for all pollution days was 0.9, which indicates that the collected PM was slightly alkaline, which is consistent with other studies [30,45]. On the other hand, the AE and CE were strongly correlated among the four pollution levels (good: $R^2 = 0.90$, light: $R^2 = 0.92$, moderate: $R^2 = 0.91$, heavy: $R^2 = 0.98$). Furthermore, the AE/CE ratios on moderate and heavy pollution days were 1.15 and 1.22 times higher than those on good days (good: 0.86, light: 0.96, moderate: 0.99, heavy: 1.05), which suggests that the particles on more polluted days were more acidic than those on good days; this is consistent with previous studies [51–53]. A high concentration of secondary inorganic ions ($NO_3^-$ and $SO_4^{2-}$) not only has a significant impact on the formation of pollutants but also could enhance the acidity of particles.

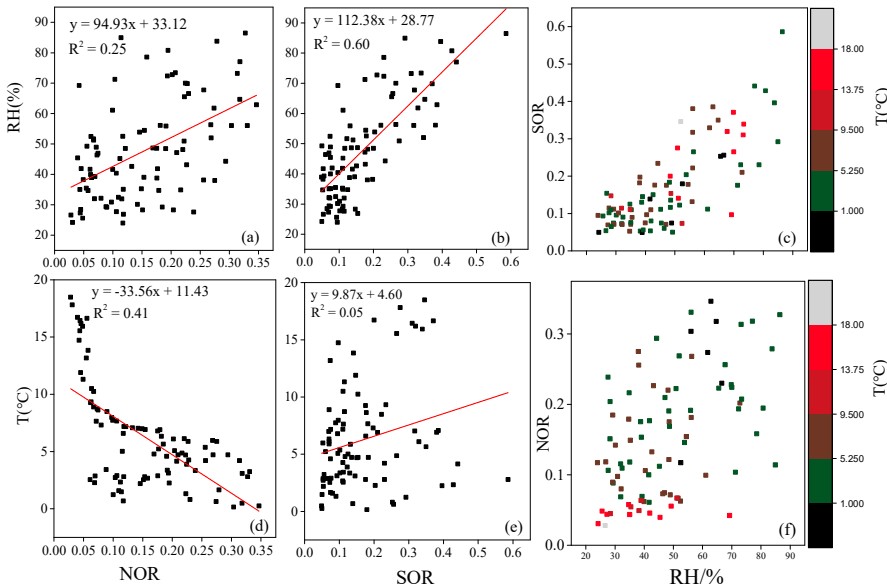

**Figure 4.** Scatter plots between RH and NOR (**a**), SOR (**b**), temperature and NOR (**d**), SOR (**e**), and the correlations of temperature and RH against SOR (**c**) and NOR (**f**) in sampling period.

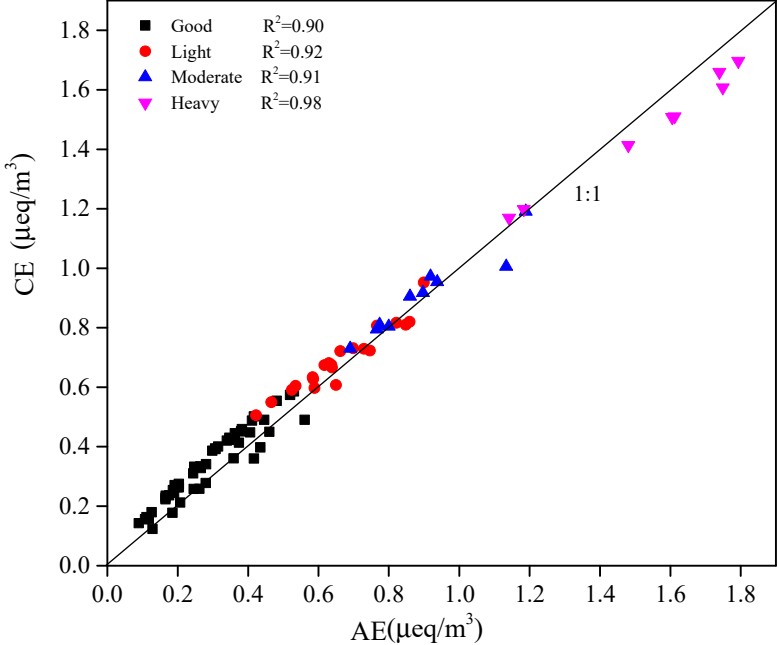

**Figure 5.** Scatter plots of AE (anion equivalent) vs. CE (cation equivalent) in four pollution days.

### 3.1.3. Carbonaceous Aerosols

The mean concentrations of OC and EC in $PM_{2.5}$ were 11.5 and 5.0 $\mu g/m^3$, accounting for 13.8% and 6.0% of $PM_{2.5}$, respectively, which is higher than Shanghai [11]. The mass concentrations of OC were higher than those of EC in each polluted level (Table 2). The proportion of OC and EC decreased from clean to polluted periods with an increase in $PM_{2.5}$ concentration (OC: 16.7% on good days and 12.5% on heavy pollution days; EC: 7.3% on good days and 5.4% on heavy pollution days), which is consistent with previous studies [54,55].

The OC/EC ratio can be used to characterize the emission and conversion characteristics of carbon aerosols [56]. In Jinan, the daily average ratio of OC/EC was 2.5, which is lower than some large cities, such as Chengdu [57], but higher than those in some industrial cities, such as Tangshan [14]

and Lanzhou [58], as shown in Table S2. In this study, the daily OC/EC ratio ranged from 1.8 to 5.7, with an average of 2.3 on good days, from 1.8 to 5.5, with an average of 2.3 on light pollution days, from 1.9 to 5.6, with an average of 2.5 on moderate pollution days, and from 1.9 to 2.8, with an average of 2.6 on heavy pollution days. The OC/EC ratios were higher on heavy pollution days than on other pollution days, and the same trends were found for $K^+$ and $Cl^-$, showing that biomass burning, fossil fuel combustion, and vehicular emissions played a significant role in the high OC/EC ratios [59,60].

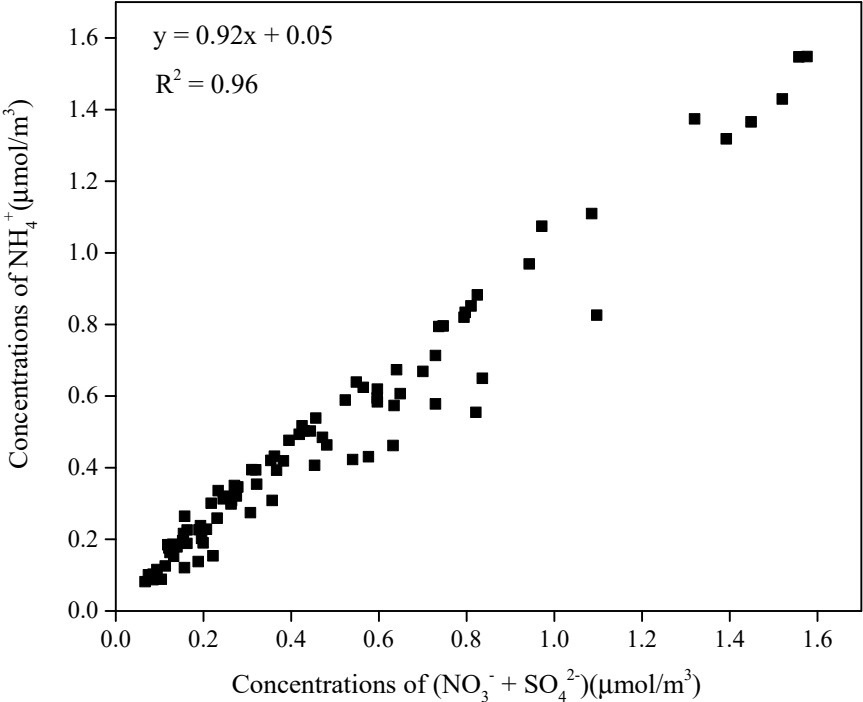

**Figure 6.** Concentrations of $NH_4^+$ versus and sum of $SO_4^{2-}$ and $NO_3^-$.

### 3.1.4. Inorganic Elements

The daily average concentration of the thirty-seven elements was 7.1 μg/m³, accounting for 8.5% of the PM$_{2.5}$ fraction in this study (Table 2). The crustal elements (Ca, Al, Na, Mg, K, and Fe) had a total concentration of 5.5 μg/m³, accounting for 77.3% of the total mass of the analyzed elements. The concentrations of the trace elements (Bi, Sn, V, Cd, Sb, Zn, Pb, Tl, Cr, Ni, As, and Cu) occupied a small part of PM$_{2.5}$. Particularly, in these trace elements, the daily average concentration of Cr was 0.02 μg/m³, exceeding the Ambient Air Quality Standards in China by 751.2 times (0.025 ng/m³, GB3095–2012).

Figure 7 presents the concentrations of the major element compositions at four pollution levels. The concentrations of the elements were higher on heavy pollution days (10.7 μg/m³) and moderate pollution days (9.4 μg/m³) than on light pollution days (6.8 μg/m³) and good days (6.0 μg/m³), and the mass concentrations of total elements accounted for 5.8% during heavy days, which were 57.5%, 19.9%, and 118.3% lower than those during good, light, and moderate episodes, respectively. Notably, the mass concentrations of most crustal elements and anthropogenic pollution elements during heavy pollution days were substantially higher than those on other pollution days. However, the proportion of all crustal elements, such as Al, Na, Mg, K, Fe, and Ca, on heavy pollution days were 63.0%, 52.6%, 55.8%, 31.2%, 59.3%, and 72.7%, lower than those on good days. These elements, which mainly originate from natural sources such as road dust or construction dust, are difficult to raise up into the atmosphere and contribute little to the PM$_{2.5}$ on heavy days due to the low wind speed and high RH.

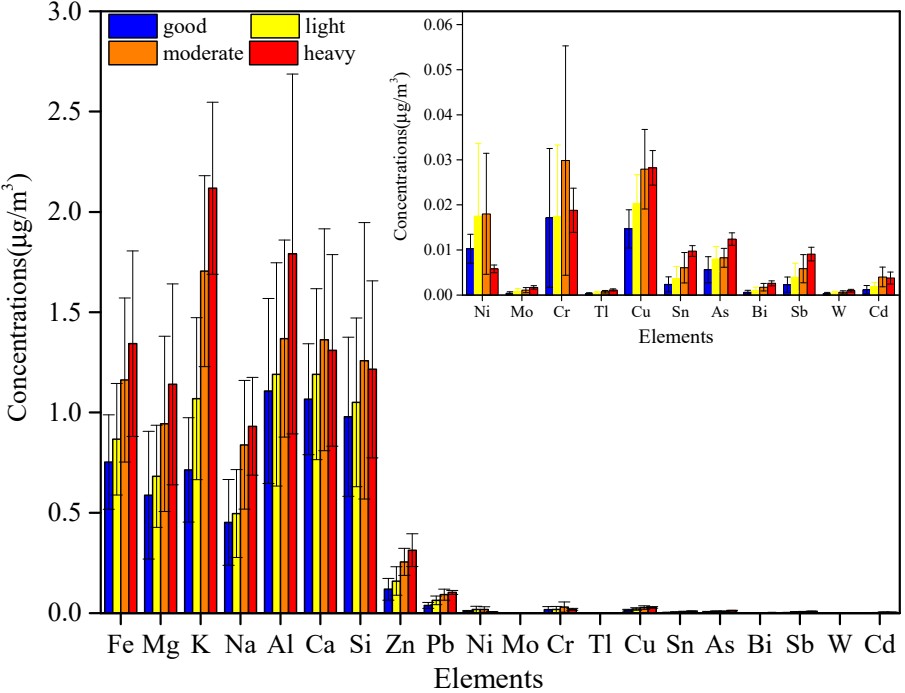

**Figure 7.** The concentrations of element compositions in four pollution days.

Figure 8 shows the EF values of elements in different pollution levels. The EF values of the elements Na, K, Al, Ca, Fe, Mg were lower than 10 during the sampling period, suggesting that these elements were mostly from mineral dusts. In contrast, the EF values for As, Cu, Cr, Tl, Pb, Sn, Zn, Bi, Sb, Si, and Cd were above 10 in all pollution levels. Moreover, the EF values were higher than 10 on good days for Ni; on light and moderate pollution days for W and Ni; and on heavy pollution days for W, As, Cu, and Mo, which suggests that they came mostly from anthropogenic sources [61]. For example, Cr, Pb, As, Cd, and Sb accumulated from coal combustion; Cu, Zn, and Tl originated from vehicle emissions; Ni was produced by oil combustion; and Sn, Bi, and Mo were associated with industrial production [62,63]. Notably, the EFs of Zn, Bi, Sb, and Cd presented the highest EF values on each pollution day, especially on heavy pollution days (188.6, 314.0, 451.7, and 1784.1), which was probably due to the coal combustion and vehicle emissions contributing to metal emissions. Overall, these elements in $PM_{2.5}$ in Jinan were mainly produced by human activities, such as coal combustion, vehicle emissions, and industry.

*3.2. Chemical Mass Closure*

$PM_{2.5}$ mass concentrations were reconstructed in four pollution levels using the method described in Section 2.3.4, compared against the measured $PM_{2.5}$ mass concentrations. The proportion of major species in $PM_{2.5}$ are shown in Figure 9. The major constituents were $NO_3^-$ and OM. Specifically, the proportions of $NO_3^-$ during the four periods was 16.4%, 23.6%, 24.1%, and 23.7%, respectively. The percentage of OM showed a change under different pollution levels, with the maximum on good days (23.2%), the minimum on heavy pollution days (17.5%), and intermediate levels on light (19.0%) and moderate pollution days (17.9%). Additionally, the proportion of mineral dust showed a marked decline; the maximum value of mineral dust was observed on good days (20.2%), while the minimum value was observed on heavy pollution days (only 7.5%). However, minimum amounts of $SO_4^{2-}$ and $NH_4^+$ were noted on heavy pollution days, accounting for 114.1% and 12.1% of $PM_{2.5}$, respectively, which increased with an aggravation of pollution. EC and TE comprise a relatively small percentage, with the maximum on good days (7.3% and 1.6%) and the minimum on heavy pollution days (5.4% and 0.8%), respectively. The contribution of unidentified constituents reached its maximum (19.0%) on

heavy pollution days and its minimum (12.3%) on good days, which needs to be further explored in future studies.

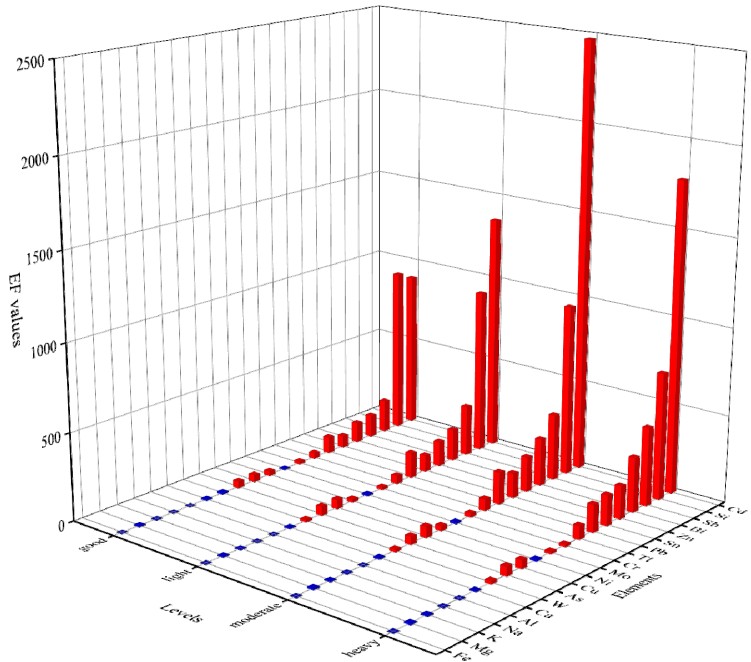

**Figure 8.** Enrichment factors of elements in four pollution days (red column ≥10, blue column < 10; The x-axis, y-axis, and z-axis represent different pollution levels, elements, and enrichment factor (EF) values, respectively).

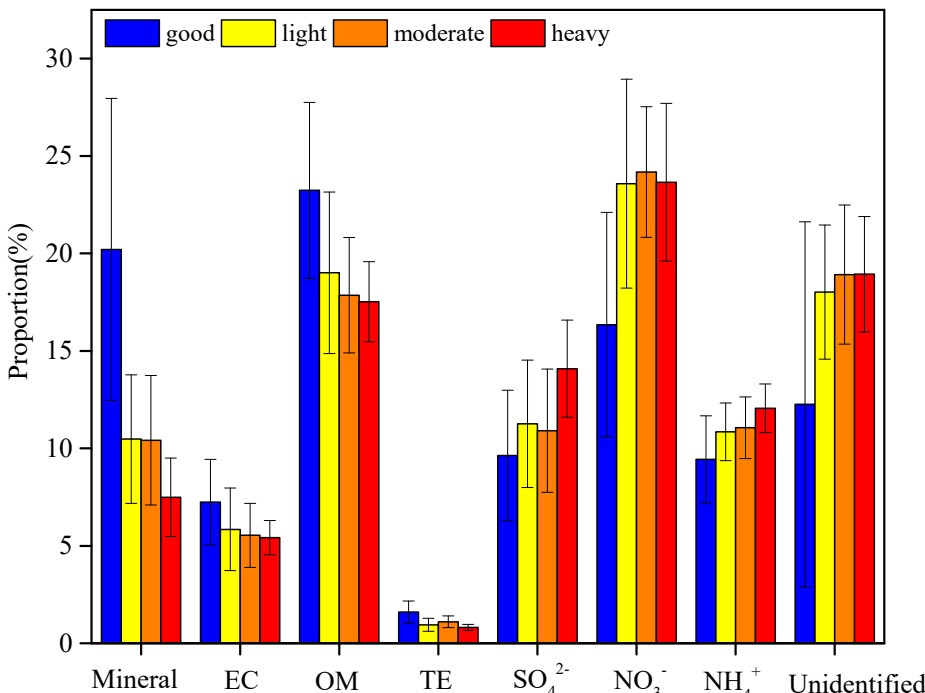

**Figure 9.** Proportion of major species in $PM_{2.5}$ in four pollution days.

*3.3. Source Apportionment Analysis*

In this study, fugitive dust, soil dust, cement dust, coal fly ash, vehicle exhaust dust, steel dust, biomass dust and secondary inorganic aerosols were determined to be the most important pollutants.

The source profile data were collected from the Environmental Protection Science Research Institute of Jinan.

Figures 10–12 present the source profiles of PM$_{2.5}$ for different types of sources. According to Figure 10, abundant amounts of Al, Si, Ca, Fe, and OC were found in mobile sources, with proportions of 8.6%, 8.0%, 5.6%, 2.2%, and 7.8% in fugitive dust samples, 8.9%, 10.2%, 6.0%, 3.8%, and 8.2% in soil dust samples, and 4.7%, 10.8%, 20.3%, 2.1%, and 5.7% in cement dust samples, respectively. The major components of coal fly ash were OC, Si, Al, and SO$_4^{2-}$, with proportions of 11.6%, 10.4%, 7.7%, and 4.7%, and the highest content of PM$_{2.5}$ in the vehicle exhaust was OC (25.8%), followed by EC (22.7%) (Figure 11). The ironmaking dust samples contained the most Fe, with proportions of 33.1% by weight, Ca and Al also comprised a certain proportion, and the chemical profile of the biomass dust in PM$_{2.5}$ was found to have major components: OC (28.5%), Cl (21.2%), K (19.1%), and EC (12.7%) (Figure 12).

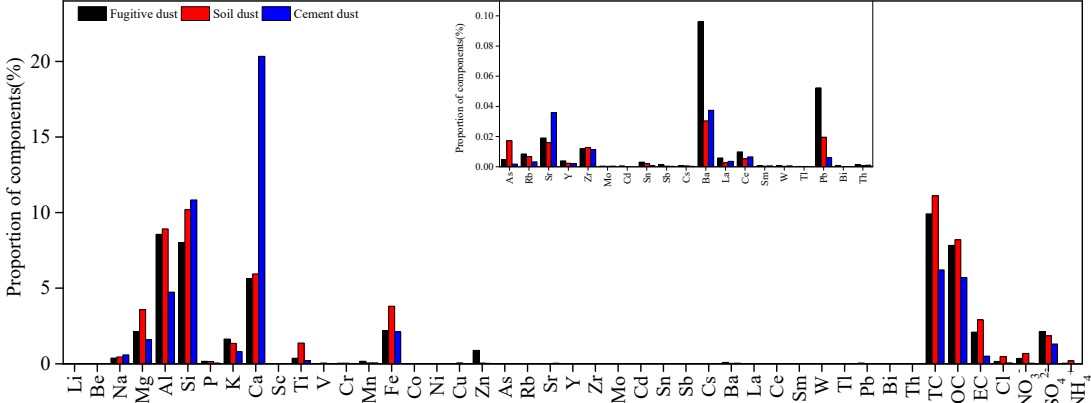

**Figure 10.** Fugitive dust and soil dust source profile.

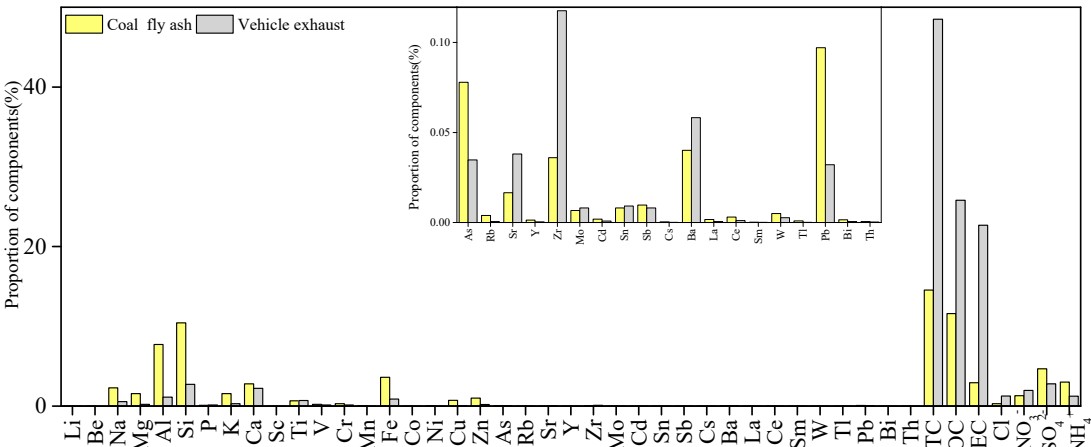

**Figure 11.** Coal fly ash and vehicle exhaust source profile.

The source apportionment results of PM$_{2.5}$ from the CMB receptor model are presented in Figure 13. Nitrate, vehicle exhaust, sulfate, and coal fly ash were the main contributors of PM$_{2.5}$ in Jinan. The contribution of nitrate was the highest in each pollution level, with 18.4% on good days, 20.5% on light pollution days, 23.4% on moderate pollution days, and 25.4% on heavy pollution days. Nitrate is mainly converted from NOx, which is derived from coal and motor vehicle exhaust. Vehicle exhaust dust was found to be the second largest contributor of PM$_{2.5}$, which showed a large difference in its contributions during the good and heavy pollution days (14.9% for good days and 18.2% for heavy pollution days). The number of motor vehicles in Jinan reached 2.06 million in 2017, an increase of 13.2% over the previous year, which also indicates that the emissions of vehicle exhaust contributed greatly to the growth of nitrogen in the atmosphere. Sulfate was the third biggest source in Jinan, the

contributions from sulfates (which are the precursors of $SO_2$ and mainly come from the combustion of coal) were 16.7% and 18.6% on moderate and heavy pollution days, respectively. Coal fly ash also contributed significantly to the $PM_{2.5}$ in Jinan, with the highest proportion on heavy pollution days (13.3%), 1.6 times higher than on good days, due to the operation of a heating boilers increasing the coal burned for heating. In addition, the contribution of open sources (fugitive dust, soil dust, and cement dust) to $PM_{2.5}$ should not be ignored, especially on good days.

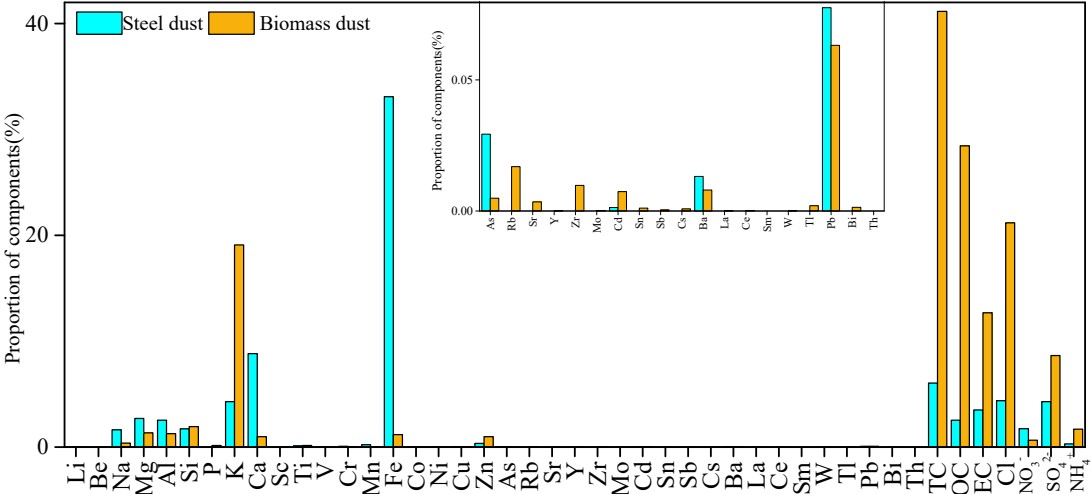

**Figure 12.** Steel dust and biomass dust source profile.

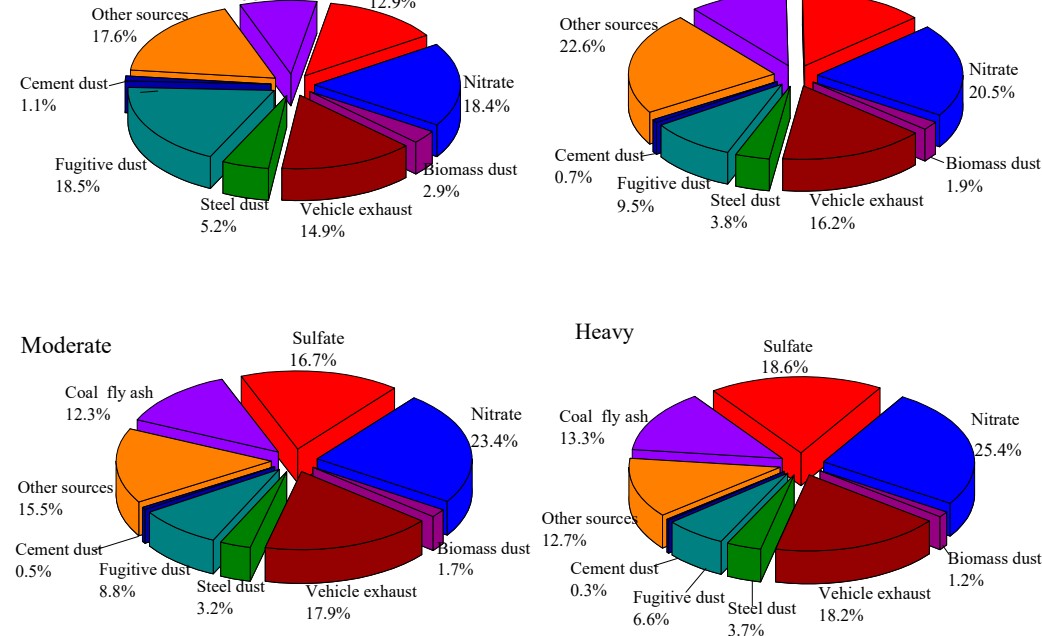

**Figure 13.** Source contribution to ambient $PM_{2.5}$ in four pollution days in Jinan.

Overall, the contribution of sulfates, nitrates, vehicle exhaust, and coal fly ash increased with an aggravation of pollution (Figure 14), while biomass dust and cement dust contributed less to $PM_{2.5}$ at each pollution level. Therefore, we suggest that the government should strengthen the control of sulfate and nitrate precursors in heavy pollution and vigorously control the emissions of coal and motor vehicle exhaust.

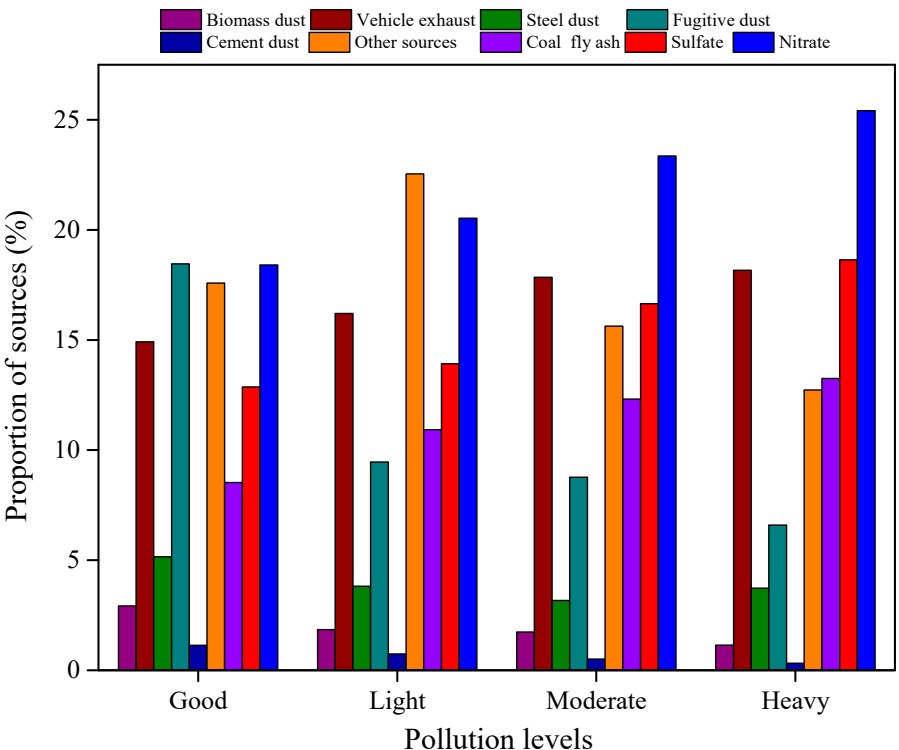

**Figure 14.** Trend of source contribution in four pollution days in Jinan.

## 4. Conclusions

Atmospheric $PM_{2.5}$ and its major components were measured from 15 October 2017 to 31 January 2018 in Jinan, China. In this study, the daily average concentration of $PM_{2.5}$ was 83.5 μg/m$^3$, which is about 1.1 times the National Ambient Air Quality Standard of $PM_{2.5}$ (75 μg/m$^3$). The concentration of $PM_{2.5}$ on heavy pollution days was approximately 4.3 times higher than that on good days, which was related to the lower wind speed and higher relative humidity on heavy pollution days. High concentrations of WSIs, carbonaceous species, and elements were observed on heavy pollution days. In addition, more $NO_3^-$, $SO_4^{2-}$, and $NH_4^+$ were formed during more polluted days, and the ion balance showed that the $PM_{2.5}$ in Jinan was slightly alkaline. The percentage of OC and EC in $PM_{2.5}$ decreased by 4.2% and 1.8% from good days to heavy pollution days, respectively. The enrichment factor showed that Zn, Bi, Sb, and Cd were contributed to by coal combustion and vehicle emission. The major constituents of Jinan were secondary aerosol particles and OM. The CMB model showed that sulfate, vehicle exhaust dust, nitrate, and coal fly ash were the main contributors of $PM_{2.5}$ in Jinan.

Although the air quality of $PM_{2.5}$ has improved slightly compared to prior years, it is still far behind the World Health Organization's recommendations and Chinese national standards. This paper studied the rapid and significant changes in the chemical composition and sources of $PM_{2.5}$ on heavy pollution days compared to on good days, revealing that motor vehicles should be better controlled to effectively reduce the concentrations on heavy pollution days.

**Supplementary Materials:** The following are available online at http://www.mdpi.com/2073-4433/11/4/336/s1, Table S1: Concentrations of $PM_{2.5}$ from Jinan and other cities. Table S2: Concentrations of $PM_{2.5}$ and major compositions from Jinan and other cities (μg/m$^3$).

**Author Contributions:** Conceptualization, S.T. and C.G.; investigation, Y.L., J.W. (Jing Wang), J.W. (Jian Wang), L.H., B.L., X.W., X.Z., and W.Y.; project administration, Z.B; software, X.W. and X.Z.; supervision, C.G., B.H., and Z.B.; writing—original draft, S.T. and C.G.; writing—review and editing, S.T. and B.H. All authors have read and agreed to the published version of the manuscript.

**Funding:** This study was funded by the Analysis of the Sources of Jinan City Scale Refinement, National Research Program for Key Issues in Air Pollution Control, China (No.DQGG0107-19), Shandong Science and Technology Development Plan Project (2014GSF117038), and Jinan Science and Technology Plan Project (201509001-2).

**Conflicts of Interest:** The authors declare no conflict of interest.

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
