# Peer review of "Chemical Compositions and Source Analysis of PM2.5 during Autumn and Winter in a Heavily Polluted City in China"

_atmosphere, doi:10.3390/atmos11040336_

Round 1

Reviewer 1 Report

General Overview:

This article explores the Atmospheric PM2.5 and its major compositions of Jinan city by conducting laboratory and modelled experiments. Please address these concerns above as well as the general statements and questions I give below as I believe that this research is of importance to the field.

General Statements:

Abstract

In general, the abstract is easy to follow. It stated the significance, objective, and the results of this study.   

Line 16-17: Details about chemical composition analysis can be discussed in the method section.

Introduction:

Line 34: No supportive information to show that the PM2.5 is composed of various carcinogenic chemicals based on reference [6,7], it claimed “Long-term exposure to combustion-related fine particulate air pollution is an important environmental risk factor for cardiopulmonary and lung cancer mortality.”, which is not equal to PM2.5 is consisted of carcinogenic chemicals. It is a wrongful statement.

Line 49-50: It is a confusing statement. It comes down to ‘the characteristics remains limited’? It is a very important transection sentence, maybe rewrite to make it easier to follow.

Line 58-60: This is the objective of this study. More importantly as the objective, it has to show how this results/information can make a impact in the relevant area. Perhaps add one or two sentences to show the potential influence introduced by this study?

 In general, this introduction is good length, precise, and easy to ready.

Materials and methods:

2.1: It is very nice to know more about the study sites and the city. However, the information needs to be focused more on the specific study/sampling sites instead of just the whole city. The entire section can be put in the introduction instead.

Line 89:  I would suggest the author to stick with the third person point writing style as the rest of this text.

Line 91-92: As stated in the text, 47mm filters were used in this study for collecting PM2.5, how has the author considered to count for the particles were smaller than that diameter range? Any estimated error analysis?

Line 93-95: Any evidence/reference to prove that this method dissolves ions sufficiently?

Line 98-99: Any reference of these detection limits? from the manufactory?

Line 101: It seems to be the first time to introduce the terms ‘OC’ and ‘EC’, please put the full names before using abbreviations.

Line 113-116: Please re-write the sentence, multiple grammar errors, it is hard to follow.

Line 118: remove ‘for’.

Line 119-120: Any reference for these detection limits?

Results and discussions:

Line 187: please also show the standard deviation if the averaged value is presented.

Table 3 & 4: They can be put in Supplementary information.

Line 226: This source information needs reference.

Figure 3: It doesn’t make any logical meanings to have the lines that are connecting different ion concentration levels. Moreover, please put error bars on your date points.

Line 233-235: Has the Pearson correlation coefficient been tested by Student's t-distribution?

Figure 6: Again, the lines between element concentration do not make sense, and, add error bars please.

Figure7: Very interesting graph, a little difficult to read though.

Figure 9: Any error bars for this proportion %?

Line 336-349: This part needs to move to the ‘Materials and method’.

Figure 14: The lines don’t make sense.

Conclusions:

It is easy to read, however, it is a summary of the result and discussion section. Please add few sentences to level this section into a real conclusion for your research.

Reviewer 2 Report

This study presents a relatively comprehensive characterization of PM2.5 chemical composition at four locations during a polluted season in Jinan, China. The study methods are marginally acceptable, and the outputs would be of interests to the readers of Atmosphere. Nevertheless, there are some points needed to be addressed before its consideration for publication as follow.

  1. In line 70, if the motor vehicles are considered one of the major sources of pollution, it would be informative to provide more details about their engine/fuel type.
  2. In line 125, it is more appropriate to use "ionic balance" than "Ion acidity" as the latter is more related to the [H+] not measured or modeled in this study.
  3. In lines 136-143, it is important that the authors make sure whether the oxidation ratios are based on molar or mass concentrations. 
  4. In the Results and Discussion, it is critical that the authors provide a rationale or logics as to why the data from four sites of different characteristics were not compared, and that the data were pooled together for discussion.
  5. In Figure 2, why are the PM2.5 conc. for good days (blue color) higher than those for other types of days? Also, not the use of significant numbers, e.g., the percentages in line 198.
  6. In Section 3.1.2, the relationships between NOR/SOR with temperature and RH should be determined more quantitatively, e.g., using regression or other methods.
  7. In Section 3.2, the authors should discuss the balance between NH4+ and (SO42-+NO3-) for neutralization, because they are the major ions. Also, the authors should clarify if the ions in mass balance were based on measured values or based on the assumption of complete neutralization.
  8. In Section 3.3, the sulfate and nitrate were considered as "individual" sources. Could the authors elaborate more about why the two were not included in the profiles of other sources?
  9. In lines 399-401, I do not find any evidence in the manuscript that supports the statement of "The ratio of NO3/SO4...."
  10. Overall, the manuscript lacks a thorough discussion with findings of other studies, but limited to a small number of China studies.
  11. This manuscript requires corrections in punctuation, spelling, and/or formatting.

Round 2

Reviewer 2 Report

The authors have adequately addressed the questions and comments raised by the reviewer. Though, there are still some minor points that must be corrected or verified before acceptance for publication, as follow.

  1. In the revised manuscript, the units of x and y axes in Fig. 6 appear to be incorrect. Please verify.
  2. The authors should check again about the use of significant figures. For example, the authors present the relative humidity down to two digits after the decimal point, i.e., hundredths (e.g., 22.09%), which is very uncommon. What is the precision of the relative humidity sensor?
  3. In line 400, the statement of "...the formation of new particles." doesn't make much sense in this context because new particle formation refers specifically about the formation of a few nanometer-sized particles of large numbers.
  4. The manuscript still requires extensive language editing by native English speakers. For example, capital letters are used in the middle of a sentence, missing spaces between numbers and units, etc.
